# CXMArena: Unified Dataset to benchmark performance in realistic CXM Scenarios

## Abstract

Large Language Models (LLMs) hold immense potential for revolutionizing Customer Experience Management (CXM), particularly in contact center operations. However, evaluating their practical utility in complex operational environments is hindered by data scarcity (due to privacy concerns) and the limitations of current benchmarks. Existing benchmarks often lack realism, failing to incorporate deep knowledge base (KB) integration, real-world noise, or critical operational tasks beyond conversational fluency. To bridge this gap, we introduce CXMArena, a novel, large-scale synthetic benchmark dataset specifically designed for evaluating AI in operational CXM contexts. Given the diversity in possible contact center features, we have developed a scalable LLM-powered pipeline that simulates the brand's CXM entities that form the foundation of our datasets — such as knowledge articles including product specifications, issue taxonomies, and contact center conversations. The entities closely represent real-world distribution because of controlled noise injection (informed by domain experts) and rigorous automated validation. Building on this, we release CXMArena, which provides dedicated benchmarks targeting five important operational tasks: Knowledge Base Refinement, Intent Prediction, Agent Quality Adherence, Article Search, and Multi-turn RAG with Integrated Tools. **Our baseline experiments underscore the benchmark's difficulty: even state of the art embedding and generation models achieve only 68% accuracy on article search, while standard embedding methods yield a low F1 score of 0.3 for knowledge base refinement, highlighting significant challenges for current models necessitating complex pipelines and solutions over conventional techniques.** CXMArena is available here: [link redacted for review].

## 1 Introduction

Customer Experience Management (CXM) systems are pivotal in modern enterprises, facilitating seamless and personalized interactions across touchpoints like voice, chat, and email (Palmer, 2010). Businesses increasingly differentiate themselves through superior customer service, and the integration of Large Language Models (LLMs) promises to revolutionize CXM by automating tasks and providing real-time support, transforming contact center operations (Sulastri, 2023).

Despite this potential, rigorously evaluating AI agents within the complexities of real-world CXM environments remains a significant hurdle. Key challenges include the scarcity of high-quality, diverse customer interaction data due to privacy concerns (Abdalla et al., 2025) and the difficulty of capturing nuanced, context-dependent interactions (Liu et al., 2022). Furthermore, existing benchmarks, while valuable for specific research goals like open-ended conversation quality (e.g., NatCS (Gung et al., 2023)) or reasoning in information-seeking dialogues (e.g., RSiCS (Beaver et al., 2020)), often fall short of reflecting the operational realities of contact centers. Critically, they frequently lack deep integration with the extensive, domain-specific knowledge bases (KBs) agents rely on, and fail to cover a range of essential operational tasks beyond core dialogue. Functions vital for efficient and effective contact center operations – such as maintaining KB accuracy (Koh et al., 2005), understanding the core reason for contact (Rekilä, 2013), ensuring interaction quality (Chen & Li, 2021), efficiently finding, and proactively assisting (Delana et al., 2020) – are often overlooked in current evaluation frameworks.

To bridge this gap, we introduce CXMArena, a novel, large-scale synthetic benchmark dataset specifically designed for comprehensive evaluation of AI in operational CXM contexts. Generated via a scalable LLM-powered pipeline, CXMArena simulates realistic, persona-driven customer-agent interactions grounded in synthesized KBs specific to a fictional business domain. Our pipeline incorporates controlled noise (e.g., simulated ASR errors, interaction fragments) to mirror real-world data variability, and extensive validation to ensure its authenticity and high quality. Crucially, CXMArena provides dedicated benchmarks specifically targeting the five essential operational tasks identified above, moving evaluation beyond conversational fluency towards practical utility.

Our contributions are as follows:

- **The Dataset:** A comprehensive, extensively validated benchmark dataset encompassing five critical CXM operational tasks often neglected by prior work. We provide the complete underlying conversational and KB data with metadata, enabling future research on related downstream tasks.
- **The Data Generation Pipeline:** A scalable pipeline to emulate persona-based generation of CXM entities such as synthesized KBs, controlled noise injection, and validation protocols. To demonstrate its versatility, we have successfully applied this pipeline to generate datasets across different domains and languages, including French (Luxury Cosmetics) and German (Smartphones), showcasing its potential to model the complexities of diverse, modern CXM environments.

## 2 Evaluating Core CXM Tasks: Rationale and Challenges

Evaluating AI for CXM requires moving beyond assessments of conversational fluency to rigorously measure performance on tasks critical to operational success. While LLMs show promise, their true value in contact centers hinges on their ability to effectively handle the core functions that drive efficiency, accuracy, and quality.

From a brand's perspective, setting up a reliable and efficient platform to improve customer experience hinges on the ability to understand, interpret and analyze the customer's concerns and convey their solutions effectively. Thus, two of the major use cases involve setting up a customer interaction system grounded in the brand's public and private knowledge corpus, as well as the ability to effectively analyze and gather insights from customer interactions. Based on this and our corporate standing in the market, we have identified five crucial benchmark tasks that help to achieve the above. The details of these are highlighted in Section 3 below.

Consider tasks heavily reliant on customer interaction and knowledge base integrity (Article Search, Multi-Turn RAG with integrated Tools, Knowledge Base Refinement). For Article Search (retrieving the right KB article), common QA datasets (e.g., SQuAD (Rajpurkar et al., 2016), HotPotQA (Yang et al., 2018), Natural Questions (Kwiatkowski et al., 2019)) typically use broad knowledge sources like Wikipedia, lacking the specific business domain context of CXM KBs and often do not guarantee the unique answer provenance needed for closed-KB retrieval evaluation. Evaluating sophisticated assistance like Multi-Turn RAG with integrated Tools (predicting the appropriate reply or performing function calls grounded in KBs during conversation) is difficult as existing dialogue or function-calling datasets (e.g., MTRAG (Katsis et al., 2025), CORAL (Nielsen et al., 2024), BFCL (Yan et al., 2024)) often lack the necessary deep integration of multi-turn context, grounded KB passage prediction, and simulated tool use within a unified conversational flow representative of real-time agent assistance. For example, tool calling may require rigorous probing on the agent's part for customer information before a step could be taken. Furthermore, for Knowledge Base Refinement (improving the KB quality), existing resources focused on textual consistency (e.g., SPICED (Wright et al., 2022) or contradiction corpora (de Marneffe et al., 2008)) typically operate at the sentence level, whereas operational KB maintenance demands identifying inconsistencies across entire articles within a domain-specific context.

Similarly, for tasks focused on conversation analytics and insights gathering (Intent taxonomy discovery and prediction, Agent Quality Adherence), current datasets present gaps. Intent taxonomy discovery and prediction (identifying the customer's main concern) requires conversation-level classification against much larger, more nuanced taxonomies (often 100+ intents) than those found in many intent detection benchmarks (e.g., the Bitext chatbot corpus (Bitext Innovations, 2024), with 27

message-level intents). While many datasets (e.g., MASSIVE (FitzGerald et al., 2022)) offer intent prediction benchmarks on messages rather than conversations, this more closely reflects real-world setting as the message which contains real customer intent is often unknown. For Agent Quality Adherence (checking agent performance against given standards), while conversational QA datasets (e.g., QAConv (Wu et al., 2022), CoQA (Reddy et al., 2019)) assess dialogue understanding, they aren't designed to evaluate adherence to the specific quality standards, compliance protocols, and interaction dynamics central to contact center quality assessments.

CXMArena directly addresses these demonstrated evaluation gaps through its deliberate design. Our synthetic generation pipeline bypasses privacy issues, enabling large-scale data creation tailored to a specific business domain. Crucially, it co-generates integrated KBs and conversations, ensuring dialogues are realistically grounded in relevant knowledge, providing verifiable links between conversational turns, KB passages, and simulated tool use – essential for robust Article Search and Multi-Turn RAG with integrated Tools evaluation in a way current resources do not. We specifically focus on article-level inconsistencies for Knowledge Base Refinement and provide conversation-level classification with larger, customizable taxonomies for Intent taxonomy discovery and prediction, moving beyond the limitations of existing intent datasets. Agent Quality Adherence tasks are formulated around explicit quality criteria embedded during simulation, offering a targeted evaluation framework missing from general conversational QA benchmarks. Furthermore, the intentional injection of controlled noise ensures the dataset better reflects the complexities of real-world interactions often absent in cleaner academic datasets.

Therefore, CXMArena provides a much-needed, integrated, and large-scale benchmark specifically designed to evaluate AI across these crucial operational CXM tasks within a more realistic and challenging environment than offered by existing resources like those mentioned above.

## 3   DATASET DESCRIPTION

Building on the operational challenges motivated in Section 2, CXMArena provides dedicated benchmark datasets designed to evaluate AI capabilities across five crucial CXM tasks. Detailed statistics of CXMArena and comparison with real world data can be found in Appendix A. Below, we outline the objective and scope of each benchmark task:

**Knowledge Base Refinement** This task assesses an AI agent's ability to maintain the integrity and quality of knowledge resources. Given a repository of KB articles, the model must perform cross-document analysis to identify article pairs exhibiting either significant semantic overlap (similarity) or conflicting factual information (contradiction). The goal is to evaluate automated methods for detecting and flagging potential inconsistencies within a domain-specific KB.

**Intent taxonomy discovery and prediction** This benchmark evaluates the AI agent's capacity to accurately discern the primary reason for a customer's contact. Given a conversation transcript and a predefined, domain-specific intent taxonomy, the model must classify the conversation according to the most fitting intent category. This task tests the model's dialogue comprehension, particularly its ability to extract the core customer need amidst conversational noise, ambiguity, or multiple discussed topics.

**Agent Quality Adherence** This task involves evaluating customer care agent performance against defined standards. The AI agent is presented with a conversation transcript along with a specific quality assessment query (e.g., 'Did the agent resolve the issue?'). The AI agent must then answer the query along with valid message IDs from the transcript supporting the answer.

**Article Search** This benchmark focuses on fundamental information retrieval accuracy within the dataset's specific business domain. Analogous to standard RAG retrieval, the task requires the AI agent to identify and return the most relevant KB article(s) from the accompanying repository in response to a direct user query. It evaluates the core ability to locate pertinent information within the closed, domain-specific knowledge base.

**Multi-turn RAG with Integrated Tools** This task evaluates an AI's capability for proactive, context-aware assistance within an ongoing dialogue. Analyzing the conversation history up to the user's latest turn, the model must anticipate the customer's requirements. The objective is to either predict

and rank the KB articles most likely needed by the agent to construct an effective next response or make the required function call.

# 4 DATASET CREATION

Our dataset is generated through an automated pipeline that simulates customer care knowledge resources and conversations. This pipeline first constructs KBs specific to a fictional brand and then uses these KBs to generate realistic conversations. For this iteration of CXMArena, we developed a fictional brand called 'GreenBuild', an innovative company specializing in sustainable smart home technology, offering products ranging from energy-efficient appliances to integrated home automation systems. This approach provides interconnected data for the various downstream tasks, as shown in Figure 1. Full implementation specifics are detailed in Appendix B.

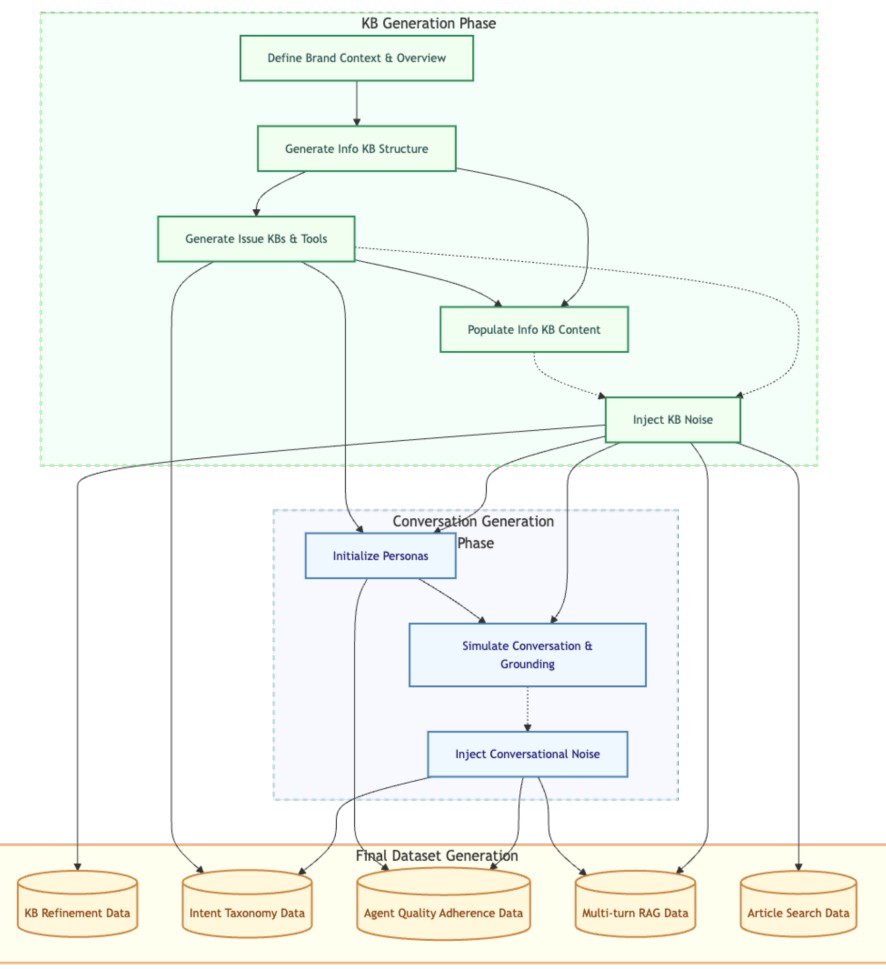

Figure 1: Workflow diagram for the CXMArena creation process, showing the steps from initial KB generation to the final extraction of benchmark task data. This is a high-level overview of our data generation pipeline. The nuanced details of KB index and content generation which ensure real-world distribution is covered in Appendix B.

## 4.1 KB GENERATION

Our methodology initiates with the automated generation of two distinct KB types specific to the defined brand context. Information KBs serve as structured repositories detailing the brand's domain

knowledge (offerings, procedures, etc.). Issue KBs highlight potential user problems pertinent to the Information KBs, along with corresponding resolution protocols. The generation involves creating a hierarchical topic structure for the Information KBs, using this structure to guide the generation of related issues and resolutions for the Issue KBs, and finally populating the Information KB structure with detailed descriptive content. One key challenge we faced was ensuring knowledge consistency across independently generated knowledge bases, which is covered in detail in Appendix B.

## 4.2 CONVERSATION GENERATION

Utilizing the generated KBs, we simulate multi-turn conversations mirroring customer-agent interactions. This involves instantiating customer and agent personas. The agent's behaviour is governed by relevant KB content, operational constraints (e.g., quality metrics), and available tools. The customer persona is based on a specific problem from an Issue KB, with variations modeling diverse user articulacy and traits. Conversations progress turn-by-turn, with agent responses grounded in traceable KB passages when factual information is provided. The simulation also incorporates agent decisions to invoke software tools as dictated by resolution paths.

## 4.3 TASK-SPECIFIC DATA DERIVATION

The generated KBs and conversations directly enable the creation of datasets for specific applications.

- **KB Refinement:** We simulate real-world data quality issues by introducing controlled redundant and contradictory information from one article to another, creating data for developing KB maintenance techniques.

- **Intent taxonomy discovery and prediction:** Each conversation is automatically labeled with the underlying customer intent derived from the Issue KB used for its generation. The complete list of possible intents forms the classification taxonomy, providing labeled data (conversation - intent).

- **Agent Quality Adherence:** Conversations are tagged with the quality parameters used during the agent persona's simulation (e.g., adherence to specific guidelines), yielding labeled data (conversation - quality assessment parameters/flags).

- **Article Search:** To create data for evaluating information retrieval, we randomly sample KB articles from the generated knowledge base and automatically generate relevant user queries using an LLM, ensuring the query's answer is primarily contained within that specific article. This process yields the necessary query-article pairs for the benchmark.

- **Multi-turn RAG:** Data for this task is derived from simulated conversations where the agent retrieved information from the KB. For instances where an agent's response was grounded in specific KB articles, we capture the conversation history up to the user's preceding turn. This history serves as the input context, and the KB articles referenced by the agent serve as the target output, creating examples for training models to predict relevant knowledge resources based on conversational context. A similar method was used for creating the tool usage dataset.

The dataset's validity and quality is ensured via an automated process, the details of which are provided for each downstream task in Appendix C. We also check the generated conversations for realism using human annotators, this process is detailed in Appendix D.

## 5 BENCHMARKS

This section presents the baseline performance evaluation for the five core tasks within CXMArena. The results establish initial performance metrics, reveal the specific difficulties inherent in these tasks, and underscore the dataset's utility for driving research in robust CXM AI solutions. Further details on the experimental setup and evaluation metrics for each benchmark can be found in Appendix E. We provide complete error and variability analysis of the benchmarks in Appendix G.

## 5.1 KNOWLEDGE BASE REFINEMENT

We first examine the KB Refinement task, specifically addressing the challenge of automatically identifying semantically similar articles. To establish baseline performance on this sub-task, we evaluated common embedding-based similarity approaches. As detailed in Table 1, the outcomes highlight the difficulty of capturing nuanced semantic overlap within CXMArena's domain-specific articles using these standard methods. Notably, while OpenAI's `text-embedding-3-large`(OpenAI, 2024) achieved reasonable precision, its capacity to identify all relevant pairs was limited, reflected in very low recall and consequently a poor overall F1-score (0.29). Other widely-used embeddings tested yielded significantly weaker results across all metrics. Note that this baseline evaluation focused solely on similarity detection, as contradiction detection typically requires different modeling approaches than the standard embedding methods assessed here.

Table 1: Baseline performance of common embedding models on the Knowledge Base Refinement task, evaluated by Precision, Recall, and F1-Score for identifying similar article pairs.

| Model | Precision | Recall | F1 Score |
|---|---|---|---|
| **text-embedding-3-large** | **0.85** | 0.18 | **0.29** |
| jina-embeddings-v3 (Sturua et al., 2024) | 0.13 | 0.23 | 0.17 |
| text-embedding-ada-002 (OpenAI, 2022) | 0.10 | 0.31 | 0.15 |
| all-mpnet-base-v2 (Song et al., 2020) | 0.16 | 0.13 | 0.15 |
| multilingual-e5-large (Wang et al., 2024) | 0.06 | **0.47** | 0.11 |

## 5.2 INTENT TAXONOMY DISCOVERY AND PREDICTION

To establish baseline performance for Intent taxonomy discovery and prediction, we evaluated models using three different taxonomies, each of which was autonomously discovered from the data using our discovery pipelines. We evaluated using direct exact matching of the predicted intent against the associated ground truth label as opposed to LLM-based verification, where an LLM determines if the prediction is correct or acceptable. We observed that the latter approach yielded very high accuracy for all three taxonomies, indicating that the intents were likely overlapping. The performance metrics resulting from these approaches are presented in Table 2.

For a fixed taxonomy, the accuracy follows the established order of model capability (GPT-4.1 (OpenAI, 2025) >GPT-4.1 mini (OpenAI, 2025) >>GPT-4.1 nano (OpenAI, 2025)) with Gemini 2.0 Flash (DeepMind, 2024) performing as well as or better than GPT-4.1 mini. We also observe that Taxonomy B is significantly higher in quality than A and C, despite having twice as many intents. This serves as an important data point for comparing taxonomy quality.

Table 2: Accuracy of baseline models on the intent prediction benchmark, evaluated across three distinct, predefined intent taxonomies (A, B, C).

| Model | Taxonomy A | Taxonomy B | Taxonomy C |
|---|---|---|---|
| **GPT-4.1** | 30.13 | **70.89** | **46.88** |
| Gemini 2.0 Flash | **30.23** | 61.90 | 45.45 |
| GPT-4.1 mini | 29.11 | 62.21 | 44.84 |
| Qwen-2.5-14B-Instruct (Qwen Team et al., 2025) | 25.64 | 49.95 | 34.83 |
| GPT-4.1 nano | 15.22 | 17.57 | 30.44 |
| Qwen-2.5-7B-Instruct (Qwen Team et al., 2025) | 20.53 | 25.84 | 28.7 |

## 5.3 AGENT QUALITY ADHERENCE

To establish baseline performance, we concentrated on the Boolean (True/False) response to quality queries. Performance was measured by directly comparing the different models' predicted True/False

answers to the ground truth labels for each query. The overall accuracy and F1 scores, presented in Table 3, reflect the average correctness across all quality assessment questions evaluated within the dataset.

One key observation is that Gemini 2.0 Flash provides performance similar to GPT-4.1 mini, whereas the 14B Qwen model shows a noticeable performance jump over its 7B and 8B counterparts. We found this trend to be consistent across multiple tasks.

Table 3: Baseline performance on the Agent Quality Adherence benchmark. Case Level Accuracy requires all questions per conversation correctly answered; Question Level Accuracy measures correctness across individual questions.

| Model | Case Level Accuracy (%) | Question Level Accuracy (%) |
|---|---|---|
| Qwen-2.5-7B-Instruct | 15.1 | 77.8 |
| Qwen-2.5-14B-Instruct | 22.5 | 82.5 |
| GPT-4.1 nano | 22.6 | 78.4 |
| Gemini 2.0 Flash | 26.3 | 83.7 |
| GPT-4.1 mini | 27.3 | 83.1 |
| **GPT-4.1** | **32.4** | **86.1** |

## 5.4 Article Search

Article Search tests the system's capabilities to fetch the most relevant top K articles given an independent user query. It is primarily used to build customer search pages and help docs for contact centers. We primarily evaluate the precision performance of different pipelines with the results being reported in Table 4. It reveals the performance using different embedding models and chunk sizes. For retrieval, faiss (Douze et al., 2024) index with L2 distance has been used.

As is evident from the table, smaller chunk size performs slightly better than large chunk sizes, but this also comes at the cost of increased number of chunks, storage space, and hence slightly more search time. Also, text-embedding-3-large performs significantly better than the openly available e5-large-instruct which is also one of the leaders of MTEB leaderboard (Enevoldsen et al., 2025).A striking finding is the significant rate of hallucination across both configurations highlighting the difficulty in strictly adhering to provided domain-specific knowledge, a crucial aspect for trustworthy CXM applications.

Table 4: Retrieval evaluation on the Article Search benchmark

| Embedding Model Used | Chunking Size | Retrieval Precision |
|---|---|---|
| **text-embedding-3-large** | 1000 | **0.75** |
| text-embedding-ada-002 + BM25 | 500 | 0.68 |
| text-embedding-ada-002 | 1000 | 0.67 |
| e5-large-instruct | 500 | 0.65 |
| e5-large-instruct | 1000 | 0.63 |
| BM25 (Robertson et al., 1995) | 500 | 0.42 |

Table 5: Qualitative evaluation of baseline RAG pipelines on the Multi Turn RAG benchmark, reporting Correctness,Incorrectnessm Hallucination rate, and Refusal rate. All the models use the same Retrieval Policy of text-embedding-ada-002:1000.

| Generator Model Used | Correct | Incorrect | Hallucination | Refusal |
|---|---|---|---|---|
| GPT-4o | 50.0 | 4.0 | 24.0 | 22.0 |
| Gemini 2.0 Flash | 61.0 | 5.0 | 13.0 | 21.0 |

## 5.5 MULTI-TURN RAG

The Multi-turn RAG benchmark evaluates an AI's capability for proactive, context-aware assistance within an ongoing dialogue, anticipating the support agent's needs based on the conversation history. This involves three key aspects assessed by CXMArena: predicting relevant Knowledge Base articles, generating appropriate responses and identifying the correct tool/function calls. First, we assess the model's ability to retrieve relevant KB information. For conversational turns where the simulation indicated a need for KB lookup, we generated a representative query based on the preceding dialogue context. Performance was measured using standard Retrieval-Augmented Generation (RAG) pipelines, evaluating Recall@topK. The baseline results, highlighting the challenges of context-aware retrieval in dialogue, are presented in Table 6.

Table 6: Performance comparison of Multi-turn RAG across different models and pipelines. The chunk size used is 500.

| Embedding Model | k | Average Recall |
|---|---|---|
| **text-embedding-ada-002** | 20 | **0.35** |
| text-embedding-3-large | 20 | 0.34 |
| text-embedding-ada-002 | 10 | 0.26 |
| text-embedding-3-large | 10 | 0.26 |
| multilingual-e5-large-instruct | 20 | 0.25 |
| multilingual-e5-large-instruct | 10 | 0.18 |

Next, we have also benchmarked the generation quality of responses using Gpt-4o as a judge ,the results of which are show in Table 5. Gemini-2.0-flash was found to be more grounded in its responses as compared to gpt-4o with lesser hallucination rate.And Lastly, we evaluate the model's ability to correctly identify the need for specific tool interactions based on the dialogue context. This is crucial for automating actions or fetching dynamic data not present in the static KB. We measured the accuracy of baseline models in selecting the appropriate tool from a predefined set when presented with conversational contexts requiring a tool call. To assess scalability, this evaluation was performed by varying the total number of available tools/functions presented to the model. Table 7 shows that the precision generally decreases as the number of potential tools increases, demonstrating the difficulty of precise tool identification in complex scenarios. **gpt-4o appears to have lower accuracy as we observed it was more talkative and often needed confirmation before making tool call. However, it depicted better precision when tool calling was forced!**

In Figure 2, we compare the average performance of different models across these analytics and embedding tasks.

Table 7: Accuracy comparison of different models on a tool/function calling task with varying numbers of available tools.

| | Number of tools/functions | | | | |
|---|---|---|---|---|---|
| Model Name | 16 | 32 | 64 | 96 | 128 |
| Gemini 1.5 Pro | 42.98 | 37.06 | 34.43 | 35.53 | 35.31 |
| Gemini 2.0 Flash | 47.59 | 45.83 | 40.57 | 41.45 | 42.32 |
| gpt-4o | 52.19 | 48.25 | 43.20 | 41.45 | 42.98 |
| gpt-4o-mini | 57.46 | 52.63 | 50.00 | 47.81 | 44.52 |

## 6 LIMITATIONS

We acknowledge several key limitations regarding CXMArena. Foremost, as a synthetically generated dataset, it may not fully replicate the complexity, unpredictability, and subtle nuances of real-world customer interactions, despite efforts to inject noise. The characteristics of the data are also inherently tied to the capabilities and potential biases of the LLMs used in its creation and validation.

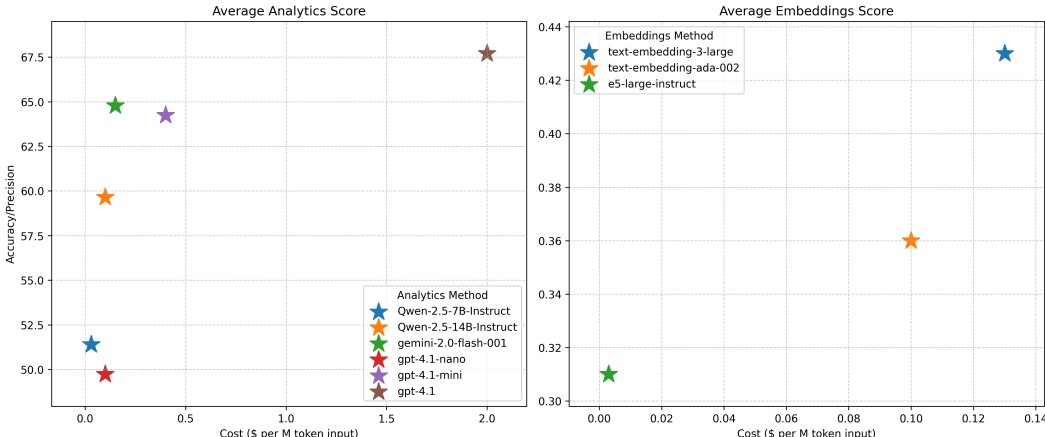

Figure 2: The first diagram shows the accuracy scores of multiple models averaged across analytics tasks, i.e, Intent Prediction and Agent Quality Adherence while the second diagrams benchmarks different embedding techniques for tasks like Article Search, Multi Turn RAG and Knowledge Base Refinement.

Furthermore, while the primary analysis in this paper is focused on a single fictional business domain within the English language, we have taken initial steps to address this limitation. To demonstrate the generalizability of our pipeline, we have generated two additional datasets: one in French for the Luxury Cosmetics domain and another in German for the Smartphones domain. Preliminary statistics and baseline results for these datasets are provided in Appendix H. While these results showcase the pipeline's adaptability, a full cross-domain and multilingual analysis remains an area for future work. We plan to formally expand the CXMArena benchmark to include other industry verticals and languages in the coming months.

## 7 CONCLUSION

We present CXMArena, a large-scale synthetic benchmark dataset designed to evaluate AI systems on realistic operational tasks found in Customer Experience Management (CXM). Current benchmarks often overlook practical contact center challenges; CXMArena fills this gap by focusing on five key areas: Knowledge Base Refinement, Intent Prediction, Agent Quality Adherence, Article Search, and Multi-turn RAG. Our baseline experiments demonstrate that these operational tasks pose significant challenges for existing models, particularly in areas such as maintaining KB integrity, accurate context-aware retrieval in dialogues, and adhering strictly to domain knowledge. CXMArena offers a much-needed tool for rigorously assessing and advancing AI capabilities beyond conversational fluency towards practical utility in CXM. By successfully extending our generation pipeline to French and German language datasets in distinct commercial domains, we have also demonstrated a clear path toward more robust and generalized multilingual CXM evaluation. **We also show that gemini-2.0-flash is an overall superior model when considering accuracy per dollar in analytical CXM tasks while openAI embeddings are still superior when retrieval tasks are concerned**. We are releasing the current dataset and associated resources to facilitate research and development.

## REPRODUCIBILITY STATEMENT

To ensure the reproducibility of our findings, we have made the CXMArena dataset publicly available on Hugging Face at [link redacted for review]. The complete source code for our data generation pipeline, as well as the scripts required to replicate the baseline benchmark evaluations presented in this paper, are accessible in our GitHub repository: [link redacted for review]. A detailed breakdown of the dataset creation process, including specifics of the models and methodologies used, is provided in Appendix B. Furthermore, Appendix E outlines the exact procedures for evaluating each of the five benchmark tasks, while Appendix A offers a comprehensive statistical analysis of the dataset

to facilitate comparison. This combination of the public dataset, open-source code, and detailed documentation is intended to allow for the full replication of our results and to encourage further research and development on the CXMArena benchmark.

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

## A  STATISTICAL ANALYSIS AND REAL DATA COMPARISON

This section provides a comprehensive statistical overview of the CXMArena, detailing the scale of the generated conversations, knowledge base, and task-specific subsets. To validate the realism of our synthetic data, we conduct a detailed quantitative comparison against an aggregated, anonymized dataset from real-world contact center operations.

Table 8 summarizes the key statistics of the generated benchmark. The dataset includes a substantial volume of conversations and knowledge base articles, which form the foundation for the five downstream tasks. The Intent Taxonomy Prediction task, for instance, is built upon three distinct taxonomies of varying size and complexity to simulate diverse classification challenges.

Table 8: Key statistics for the CXMArena benchmark dataset, detailing overall data size and composition for conversations, knowledge base, and task-specific subsets.

| Category | Metric | Value |
|---|---|---|
| **General Conversations** | Total Simulated Conversations | 1994 |
| **General Knowledge Base** | Information KB Articles | 1743 |
|  | Issue KB Articles | 1425 |
| **Task-Specific Data** | | |
| KB Refinement | Total KB Articles | 1915 |
|  | Annotated Similar Article Pairs | 518 |
|  | Annotated Contradictory Article Pairs | 293 |
| Intent Taxonomy Prediction | Labeled Conversations | 979 |
|  | Taxonomy A (Intents / Levels) | 95 / 1 |
|  | Taxonomy B (Intents / Levels) | 208 / 2 |
|  | Taxonomy C (Intents / Levels) | 37 / 1 |
| Agent Quality Adherence | Labeled Conversations | 1987 |
|  | Average Queries Per Conversation | 9 |
| Article Search | Searchable KB Articles (Information KB) | 1743 |
|  | Generated Search Queries | 797 |
| Multi-turn RAG | Labeled Conversations | 566 |
|  | Target KB Articles | 3381 |
|  | Total Tools Available | 150 |

To further assess the fidelity of our generation process, we performed a comparative analysis between CXMArena and real-world operational data, averaged across multiple CXM clients. The tokens for this analysis were calculated using the 'cl100k_base' tokenizer. Key comparative findings are summarized in Table 9.

The comparative analysis demonstrates that CXMArena closely mirrors the statistical properties of real-world CXM data. Lexical diversity metrics and overall vocabulary size are nearly identical, indicating a comparable level of linguistic complexity. The structure of conversations, particularly the number of turns and turn-taking dynamics, also shows strong alignment. While synthetic messages are slightly longer on average, their distribution is comparable to authentic interactions. Similarly, the average length of synthetic KB articles closely matches that of real-world documentation, suggesting our generation process effectively captures the scale and complexity required for realistic information retrieval tasks. Since all downstream tasks are constructed directly from these core conversations and knowledge bases, these statistics provide a comprehensive validation of the realism of the entire benchmark. These findings strongly support the suitability of CXMArena as a representative and high-fidelity benchmark for CXM research.

Table 9: Comparative statistics between the synthetic CXMArena and aggregated real-world CXM data.

| Metric | CXMArena (Synthetic) | Real-World Data |
|---|---|---|
| **Lexical Diversity** | | |
| Vocabulary Size | 18,720 | 18,710 |
| Type-Token Ratio (TTR) | 0.006 | 0.008 |
| Moving-Average TTR (MATTR) | 0.747 | 0.754 |
| **Conversation Structure** | | |
| Turns per Conversation (Mean) | 44.0 | 47.1 |
| Turns per Conversation (Median) | 39.0 | 39.0 |
| Consecutive Turns per Speaker (Mean) | $\sim$1.0 | $\sim$1.0 |
| Consecutive Turns per Speaker (Max) | 3 | 8 |
| **Length Statistics (Tokens)** | | |
| Message Length (Mean) | 21.9 | 18.5 |
| Message Length (Median) | 19.0 | 14.0 |
| Total Tokens per Conversation (Mean) | 971.9 | 1335.1 |
| Total Tokens per Conversation (Median) | 918.0 | 1155.0 |
| KB Article Length (Mean) | 1289.9 | 1031.1 |

## B  DATASET CREATION PROCESS

This appendix provides a detailed description of the pipeline used to generate the synthetic dataset, as detailed in section 3. The main model used for data generation is Google's **Gemini 2.0 Flash 001**.

### B.1  KB GENERATION

Our pipeline starts with generation of KBs. For our downstream tasks, we require two types of KBs: the information KB – which highlights the offerings and operations of the brand, and the Issue KB – which highlights the potential issues faced by the customer and their potential resolutions. The KB generation process is structured as follows:

#### B.1.1  BRAND CONTEXT DEFINITION AND OVERVIEW GENERATION

The process begins with manual input, defining the target brand's industry vertical and key characteristics. Using this seed information, an LLM is prompted to generate a comprehensive brand overview narrative. This narrative elaborates on the brand's core identity, primary products and services, operational scope, and unique selling propositions (USPs), serving as the foundational context for all subsequent automated generation steps.

For this iteration of CXMArena, we begin with a target brand called 'GreenBuild'. GreenBuild is conceptualized as an innovative company operating in the sustainable smart home technology sector. Its offerings include a diverse range of energy-efficient smart appliances (e.g., smart thermostats, solar panels, smart lighting systems), home automation solutions, and associated subscription services (e.g., security monitoring, energy management platforms). This definition outlines GreenBuild's core identity, primary products and services, operational scope, and unique selling propositions, serving as the foundational context for all subsequent automated generation steps.

#### B.1.2  KB HIERARCHICAL STRUCTURE GENERATION

Leveraging the brand overview, we proceed to define the organizational structure for the KBs. An LLM constructs a hierarchical topic tree. This involves:

- **Predefined Root Nodes:** Establishing fixed top-level categories (e.g., `product_catalog`, `service_offerings`, `membership_programs`, `company_policies`) to ensure foundational coverage.

- **LLM-driven Expansion:** Prompting the LLM to expand these root nodes into a multi-level tree structure based on the Brand Overview. For instance, `product_catalog` might branch into `electronics -> mobile_phones -> Model_X`, with further nodes for components like `Model_X_battery` or `Model_X_screen`. The depth of these is decided by the LLM and varies from case to case.

This resulting tree defines the schema or index for the KBs, outlining the topics to be covered and their relationships, but does not yet contain the detailed content.

### B.1.3 KB CONTENT GENERATION AND ORGANIZATION

Using the index structure defined in the KB hierarchical structure as reference points, we generate the content for KBs. This involves several sub-steps:

- **Property Tagging:** As the index was being generated, we also generated a dictionary of properties as key-value pairs, which was used to ensure knowledge remained consistent across multiple articles. These property tags were populated as the index expanded.

- **Information Population:** The LLM is prompted to generate the article content and is seeded with metadata like property tags, content length, and content type. For each KB in the hierarchy, an LLM generates a comprehensive description of the topic represented by the node, ensuring:

    - **Metadata:** The LLM ensures metadata keys are applied consistently across sibling nodes (nodes at the same level with the same parent), while generating distinct, contextually appropriate values for each specific node.
    - **Interdependencies:** The LLM identifies and encodes functional relationships between sibling nodes where applicable (e.g., specifying that the price of a product variant depends on its size attribute).

### B.1.4 FUNCTION TOOL INTEGRATION WITHIN ISSUE KBS

For Issue KB articles where the resolution path requires executing a specific action (e.g., rescheduling a delivery, processing a refund request, checking account status), corresponding function-calling tool definitions are integrated. These definitions specify the tool's purpose, required parameters, and expected output, effectively representing an API endpoint or backend function available to a support agent. These are embedded directly within the relevant resolution steps of the Issue KBs.

### B.1.5 STRUCTURAL AND LINGUISTIC NOISE INJECTION IN KBS

The KB generation process initially yields articles in a clean, canonical format. To better approximate the imperfect characteristic of real-world knowledge repositories, we subsequently introduce controlled noise through several transformation steps:

1. **Structural Formatting Simulation:** We alter the textual presentation of KB articles to mimic various common formats, without necessarily changing the underlying file type. This involves:

    (a) Table Transformation: Identifying suitable structured information within the text and reformatting this data into textual table representations.
    (b) HTML Structure Simulation: Injecting HTML tags (e.g., `<h1>`, `<p>`, `<ul>`, `<li>`) into the text to represent common web-based KB layouts.
    (c) Markdown Conversion: Reformatting sections using Markdown syntax (e.g., `#` for headers, `*` for list items, `---` for separators).
    (d) PDF Layout: We convert the documents into PDFs using simple Python libraries, ensuring diversity in the structuring and layout of the KBs.

2. **Linguistic Noise via Acronym Introduction:** To simulate the common use of abbreviated forms, we systematically identify and introduce relevant acronyms based on the generated KB content. This sub-process involves three distinct stages:

(a) Candidate Phrase Extraction using N-grams: We first analyze the entire KB corpus to identify potential multi-word expressions that might have standard acronyms. This is achieved by extracting frequent n-grams (typically for n=2, 3, and 4 words). We apply heuristics to filter these n-grams, primarily selecting those where all constituent words are capitalized, as this pattern often indicates a formal name or term amenable to acronymization (e.g., "Graphics Processing Unit", "Customer Relationship Management"). This stage yields a broad set of candidate phrases.

(b) LLM-based Filtering and Acronym Validation: The raw list of candidate phrases inevitably contains entries that are capitalized but do not have common acronyms. We provide this list of candidate acronyms to GPT-4o to extract potential acronyms, including common ones and those that could potentially be created given the brand's context.

(c) Probabilistic Acronym Substitution: Finally, we perform a text substitution pass over the KB articles using the validated phrase-acronym mapping. To mimic natural language variation where both full forms and acronyms are often used, we employ a probabilistic approach to substituting these acronyms in the KB repository.

## B.2    CONVERSATION GENERATION

Realistic customer care conversations are simulated using the generated KBs as grounding truth.

### B.2.1    PERSONA INITIALIZATION

- **Agent Persona:** Generated using the Information KB, the relevant Issue KB (mapping the customer's problem), predefined Quality Management parameters, and available function-calling tools. The agent is primed to use KB information and tools appropriately.

- **Customer Persona:** Derived from a specific Issue KB, outlining the customer's problem. Contextual noise is introduced by using an LLM to identify and mask less critical pieces of information in the initial problem description, simulating vague or incomplete customer input. Randomly assigned personality traits (e.g., polite, impatient, confused) influence the customer's language and interaction style.

Metadata capturing the specific KBs, persona parameters (including quality metrics and tools), and customer context used for generating each dialogue is stored alongside the conversation, as illustrated in Figure 3.

### B.2.2    TURN-BASED SIMULATION

- The conversation proceeds in turns, typically initiated by the customer stating their issue (influenced by their persona and noisy context).

- The agent responds based on its persona, quality adherence goals, and the conversation history. Responses can be general knowledge, information directly sourced from the KBs, or involve tool interaction (either gathering information needed for a tool call, like a booking ID, or executing a tool function).

- **Grounding:** For every agent turn relying on specific knowledge, a filtering and ranking mechanism identifies the most relevant KB excerpt supporting the response. This ensures traceability to a ground truth source.

- **Tool Interaction Logging:** When tools are used (for information gathering or action execution), the event, including any parameters passed or data retrieved, is logged in the conversation's metadata.

Figure 4 illustrates the message-level metadata that records KB grounding and tool usage for specific turns within the conversation.

### B.2.3    SIMULATING REAL-WORLD CONVERSATIONAL NOISE

The purely generated conversational flow is intentionally perturbed to better approximate the complexities and imperfections of live customer service dialogues. This involves the injection of the following noise categories:

```
KBs:
  ▪ Shipping Restrictions
  ▪ Shipping & Delivery Policy -> Shipping Methods -> Local Delivery
  ▪ Product Range -> Eco-Friendly Insulation -> Specialty Insulation -> Bio-Based Foam Insulation -> Material Composition
  ▪ Issues -> R-Value Info Inconsistency
  ▪ Issues -> Formaldehyde Labeling Concerns
  ▪ Shipping & Delivery Policy -> Shipping Methods
  ▪ Shipping & Delivery Policy
  ▪ Product Range -> Eco-Friendly Insulation -> Specialty Insulation -> Bio-Based Foam Insulation
  ▪ Product Range -> Eco-Friendly Insulation -> Specialty Insulation
  ▪ Product Range -> Eco-Friendly Insulation
  ▪ Product Range

Skills:
  ▪ Technical Specialist Connect
  ▪ Local Supplier Lookup
  ▪ Third-Party Update Request
  ▪ Escalate Concerns
  ▪ Installer Connect
  ▪ Product Issue Escalation
  ▪ Carbon Offset Contribution
  ▪ Financing Options Assistance
  ▪ Product Replacement
  ▪ Indoor Air Quality Assessment
  ▪ USPS Tracking Guidance
  ▪ Share Specification Links
  ▪ Report Content Errors
  ▪ SDS Review
  ▪ Website Improvement Suggestions
  ▪ VR Installation Simulations
  ▪ SDS/MSDS Request
  ▪ Shipping Claim Filing
  ▪ Manufacturer COA Verification
  ▪ Compatibility Assessment Tool
  ▪ Product Management Consultation
  ▪ Professional Removal Service

Quality adherence parameters:
  ▪ Did the agent ask customer about their day at the beginning of the conversation: Yes
  ▪ Was the agent friendly and warm: Yes
```

Figure 3: Example of conversation-level metadata associated with a simulated dialogue in CXMArena, detailing the Knowledge Bases (KBs), agent persona parameters (e.g., quality metrics, tools/skills), and customer context used during generation.

```
Speaker: agent
  Text: Local delivery may incur remote area surcharges even within the 50-mile radius, which are displayed during checkout.
  Message Class: KNOWLEDGE_BASE
  Message Source: Shipping & Delivery Policy -> Shipping Methods -> Local Delivery
  Knowledge Base Relevance: Shipping & Delivery Policy -> Shipping Methods -> Local Delivery, Shipping & Delivery Policy -> Shipping Methods, Shipping & Delivery Policy, Shipping Restrictions

Speaker: agent
  Text: Remote area surcharges might apply due to logistical challenges, even if not explicitly listed. Your location might still be considered remote for delivery purposes.
  Message Class: KNOWLEDGE_BASE
  Message Source: Shipping Restrictions
  Knowledge Base Relevance: Shipping & Delivery Policy -> Shipping Methods -> Local Delivery, Shipping & Delivery Policy -> Shipping Methods, Shipping & Delivery Policy, Shipping Restrictions

Speaker: agent
  Text: Surcharges can apply to certain addresses within the 50-mile radius, possibly due to remote access challenges, not directly related to the electric vehicle fleet's capacity.
  Message Class: KNOWLEDGE_BASE
  Message Source: Shipping & Delivery Policy -> Shipping Methods
  Knowledge Base Relevance: Shipping & Delivery Policy -> Shipping Methods -> Local Delivery, Shipping & Delivery Policy -> Shipping Methods, Shipping & Delivery Policy, Shipping Restrictions

Speaker: agent
  Text: I understand your frustration. We can explore offering a refund for the shipping cost or a discount on future purchases to address this issue.
  Message Class: KNOWLEDGE_BASE
  Message Source: Shipping Restrictions
  Knowledge Base Relevance: Shipping & Delivery Policy -> Shipping Methods -> Local Delivery, Shipping & Delivery Policy, Shipping & Delivery Policy -> Shipping Methods, Shipping Restrictions
```

Figure 4: Illustration of message-level metadata within CXMArena, showing how individual agent messages are grounded to specific Knowledge Base (KB) passages or linked to tool usage events.

- **Simulated Fragmentation:** To account for hesitant speakers, interruptions, or artefacts from transcription processes, user utterances undergo probabilistic fragmentation. Syntactic rules and random chance determine potential split points (e.g., after clauses, at punctuation, or mid-phrase) to break longer utterances into shorter, potentially less fluent segments.

- **Simulated ASR Imperfections:** Recognizing the prevalence of voice channels, textual representations of user messages are modified to simulate errors typical of Automatic Speech Recognition (ASR) systems. This simulation is performed by leveraging OpenAI's GPT-4o. The model is prompted to rewrite the original user text in a way that incorporates plausible ASR-like errors, such as phonetic misinterpretations, word substitutions, deletions, or insertions, thus mimicking the output of a real-world ASR system.

- **Inclusion of Interactive Voice Response (IVR) Interaction:** To model common call center entry points, conversations are frequently initiated with a simulated IVR exchange. This automated front-end involves several turns mimicking basic menu navigation, intent gathering prompts, or authentication requests before the simulated 'human' agent interaction begins, reflecting a standard multi-stage customer journey.

An example of a complete simulated conversation transcript resulting from this process is shown in Figure 5.

```
IVR_Bot: Welcome to Green Build Depot customer support. For order inquiries, press 1. For information on our eco-friendly delivery options, press 2. For all other inquiries, press 3.
User: 1
IVR_Bot: You have selected order inquiries. If your inquiry is related to a delivery fee, press 1. If your inquiry is related to tracking your order, press 2. For all other order inquiries, press 3.
User: 1
IVR_Bot: Thank you. To better assist you with your delivery fee inquiry, please confirm the last four digits of the order number.
User: (omitted for privacy)
IVR_Bot: Thank you. Please hold while I connect you with a customer support representative who can assist you further.
Agent: Thanks for contacting Green Build Depot Customer support, I understand you have a question about a delivery fee on your order, how may I help you?
User: hey there I was just charged a $ seventy five doller delivery fee for my
User: order today and um I am confused because I thought lo cal delivery was free can you help clarify this
Agent: Local delivery may incur remote area surcharges even within the 50-mile radius, which are displayed during checkout.
User: I checked the website, and my
User: zip code isn't listed in any remote area surcharge list.
User: Can you explain why I was charged?
Agent: Remote area surcharges might apply due to logistical challenges, even if not explicitly listed. Your location might still be considered remote for delivery purposes.
User: That's strange.
User: I'm only 45 miles from
User: the store, and it didn't mention any surcharges during checkout.
User: Could it be related to the electric vehicle fleet's capacity?
Agent: Surcharges can apply to certain addresses within the 50-mile radius, possibly due to remote access challenges, not directly related to the electric vehicle fleet's capacity.
User: i see its just at the fee wasnt mentioned during check out which is frus trating is there any
User: chance of a refund or dis count given the confusion
Agent: I understand your frustration. We can explore offering a refund for the shipping cost or a discount on future purchases to address this issue.
User: a refund for the shipping cost would re ally help I chose green build de po for its echo friendly deli very and this fee caught me off guard thanks for under
User: standing
```

Figure 5: Example of a simulated customer-agent conversation generated for the CXMArena dataset.

### B.3 KNOWLEDGE BASE REFINEMENT DATA

To create data for the KB refinement task, we simulate common inconsistencies found in real KBs:

- **Simulating Redundancy:** We introduce redundant information in two ways. Firstly, entire KB articles are sometimes duplicated. Secondly, specific high-relevance information segments (identified via metadata) are extracted from one article and purposefully inserted into a related article (e.g., a sibling or parent in the KB tree) to create partial semantic overlap.

- **Simulating Contradictions:** To create contradictions, key factual elements (located using metadata, like prices or policy details) are extracted from an article using the article's metadata. Using GPT-4o, we then modify this information to generate a factual mismatch. This altered, contradictory statement is subsequently inserted into a different, randomly chosen sibling KB article. This process is depicted in Figure 6.

### B.4 INTENT TAXONOMY DISCOVERY AND PREDICTION DATA

Data generation for Intent Taxonomy Discovery and Orediction follows a multi-stage process designed to create a realistic and refined taxonomy based on both the initial problem definitions and their manifestation in simulated dialogues.

- **Initial Taxonomy Creation from Issue KBs:** The process begins with the collection of generated Issue KBs (described in A.1.3). These Issue KBs, representing potential customer

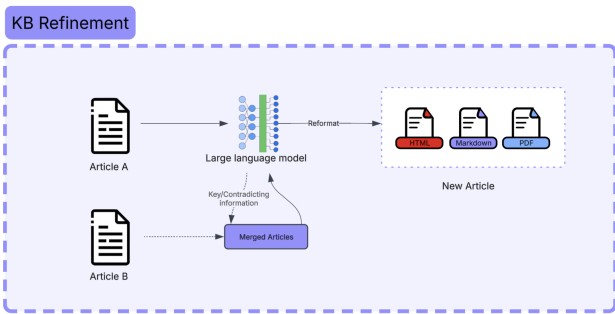

Figure 6: Illustration of the data generation for the Knowledge Base Refinement task. Clean KB articles are processed to introduce controlled redundancy (similarity) or factual inconsistencies (contradiction), creating labeled pairs for evaluation.

        problems, undergo an iterative clustering process. This step groups semantically similar issues together, forming an initial, draft taxonomy based purely on the defined problems.

- **Conversation Generation and Intent Detection:** Conversations are then generated, typically seeded by specific issues from the Issue KBs (as described in A.2). After generation, an intent detection model or process is applied to each complete conversation transcript to identify the primary intent expressed within that specific dialogue context.

- **Taxonomy Refinement and Final Labeling:** The detected intents from the conversations, potentially along with other conversational features, are then subjected to a second round of iterative clustering. This step refines the initial taxonomy by considering how intents are actually expressed and potentially co-occur in realistic dialogues. This may involve merging closely related intents, splitting broad categories, or adjusting the hierarchy based on the conversational data.

- **Ground Truth Assignment:** Finally, each conversation transcript is assigned the intent label derived from the detection step (Step 2), but mapped onto the *final, refined taxonomy* resulting from Step 3. This provides the ground truth data pair (conversation transcript, final intent label) used for the benchmark task.

This iterative process aims to produce a more operationally relevant taxonomy than one derived solely from predefined issues. The conceptual flow of this data generation is illustrated in Figure 7.

### B.5 AGENT QUALITY ADHERENCE DATA

Data for the quality adherence task is also an inherent output of the conversation simulation. As described in Section A.2, step 1, the agent persona is generated with specific quality adherence parameters governing its behaviour (e.g., adherence to script, empathy level, efficiency target). This quality adherence parameter set, used during the conversation's generation, is stored as metadata associated with that conversation. This creates a direct mapping between the conversation content/flow and the underlying quality adherence objectives or metrics, providing labeled data for training or evaluating quality adherence systems. Figure 8 shows this process.

### B.6 ARTICLE SEARCH DATA

For this task, we pick only the leaf nodes of the Information KB tree, this ensures that the selected KBs are majorly unrelated in the information they hold. Then using these KBs, we generate a query that can only be answered by the information present in the corresponding KB. This gives us the query – KB pair that is required for the Article Search task, as shown in figure 9.

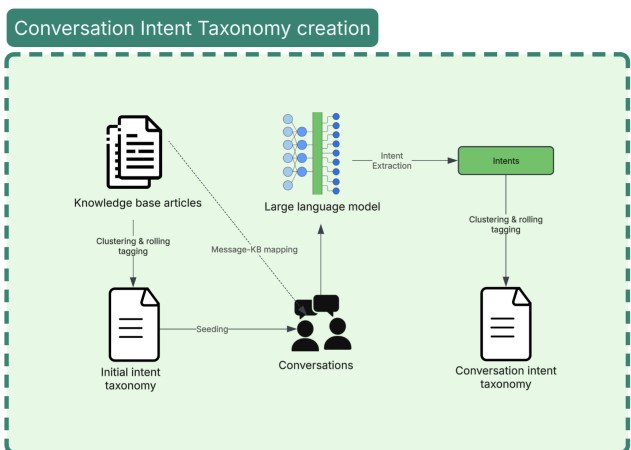

Figure 7: Data derivation for the Intent Prediction task. Each simulated conversation is automatically linked to the specific Issue KB article (representing the core customer intent) used during its generation, providing a ground truth label.

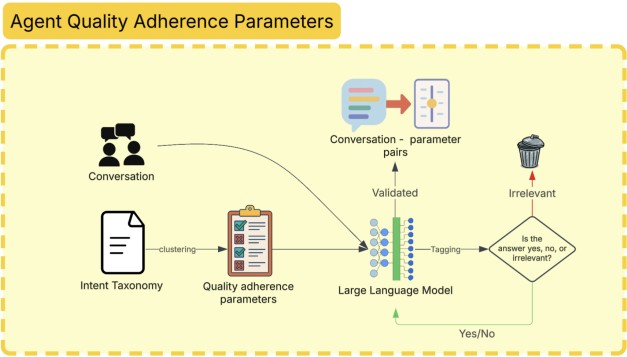

Figure 8: Generating data for the Agent Quality Adherence task. The quality parameters defined for the agent persona during simulation are stored as metadata, creating labeled examples (conversation, quality assessment query/flag) for evaluation.

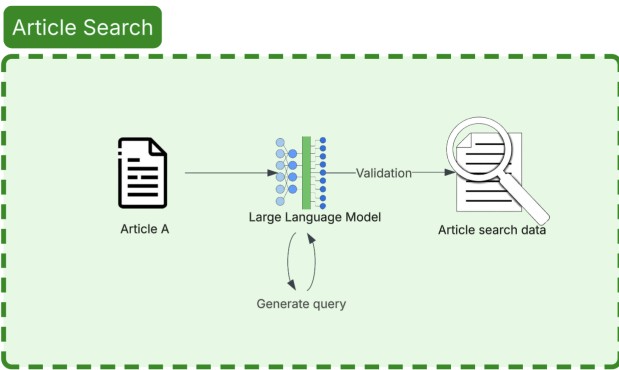

Figure 9: Process for creating Article Search data. An Information KB article is selected, and an LLM generates a relevant query whose answer is contained primarily within that specific article, resulting in a query-article pair.

### B.7 MULTI-TURN RAG DATA

The dataset for the Multi-turn RAG task is created by extracting specific examples from the full conversation simulations described in Section A.2. The key is leveraging the "KB grounding" information recorded during simulation, which links agent responses to the specific KB passages used to generate them. The process is as follows:

1. **Identify Grounded Turns:** Within each simulated conversation, we identify all agent turns where the response was explicitly based on information retrieved from the Knowledge Base. These turns have associated grounding metadata pointing to the source KB passage(s).

2. **Select Example Turn:** For each conversation containing at least one grounded agent turn, we randomly select one such turn to create a data point. We then validate this turn to ensure the agent truly needs factual information from the KB(s) to respond appropriately.

3. **Define Input Context:** We take the entire conversation history up to and including the user's message that immediately precedes the selected grounded agent turn. This sequence forms the input context.

4. **Define Target Output:** The target output consists of the identifiers (and any associated relevance data, like scores) of the specific KB passage(s) that were used to generate the agent's response in the selected turn.

This procedure transforms the simulation logs into structured pairs. These pairs directly support the training and evaluation of models designed to predict the necessary knowledge resources an agent needs based on the ongoing dialogue context and the user's latest request. This is diagrammatically represented in figure 6.

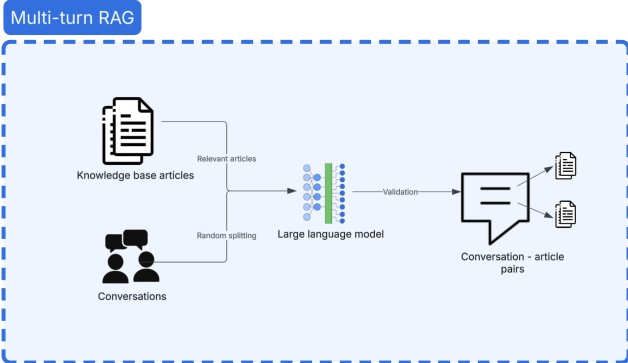

Figure 10: Derivation of Multi-turn RAG data. For agent turns grounded in the KB, the preceding conversation history (input context) is extracted and paired with the specific KB passage(s) used by the agent (target output).

## C DATASET VALIDATION PROCESS

To ensure the quality and utility of the generated benchmark datasets, we implemented task-specific validation procedures primarily leveraging LLMs for assessment. This appendix details the validation methods applied to key components of the CXMArena.

### C.1 INTENT TAXONOMY DISCOVERY AND PREDICTION

The integrity of the conversation-to-intent mappings generated for the intent recognition task was verified using an LLM. For a sample of the generated data, each conversation transcript along with its assigned ground-truth intent label (derived from the originating Issue KB) was presented to an LLM. The model was tasked to assess whether the conversational content provided sufficient semantic evidence to justify the assigned intent label, considering the context of the full intent taxonomy. This

validation confirmed that the assigned intent was appropriate and contextually supported in 98% of the evaluated cases.

## C.2 ARTICLE SEARCH

The quality of the query-article pairs created for the Article Search benchmark was evaluated for answerability. Each generated user query and its corresponding target Knowledge Base article were provided to an LLM. The LLM's objective was to determine if the information contained solely within the provided article was sufficient to comprehensively answer the query. This validation process indicated that 98% of the generated queries could be fully addressed using only the content of their designated KB article, confirming the self-contained nature of the task instances.

## C.3 MULTI-TURN RAG

Validation for the Multi-turn RAG data focused on the relevance of the ground-truth KB passages identified during simulation. For each data instance (comprising the conversation history up to the point of information need and the target KB passage for each message), an LLM assessed the semantic relevance between the dialogue context and the content of the designated KB passage. The goal was to confirm that the passage identified during generation was indeed pertinent for the agent to formulate the next response. This check affirmed the relevance of the grounded KB passage in 99% of the evaluated instances, indicating high fidelity in the automatic grounding process.

## C.4 AGENT QUALITY ADHERENCE

The automatically generated labels for the Agent Quality Adherence task underwent a two-stage validation procedure to enhance reliability.

1. **Initial Assessment and Confidence Scoring:** An LLM first reviewed the conversation transcript against the specific quality adherence query, independently determining the Boolean answer (True/False) and providing a confidence score for its judgment.

2. **Adjudication for Low Confidence:** Cases where the initial LLM reported low confidence were subjected to a secondary review. A separate LLM instance, acting as a 'judge', was provided with the original conversation, the quality adherence query, and the initial LLM's answer and justification. This judge LLM performed a final evaluation to confirm or correct the initial assessment.

This multi-step process was designed to improve the accuracy of the quality adherence labels, particularly for more nuanced or borderline cases.

## C.5 KNOWLEDGE BASE REFINEMENT

Direct external validation was not performed for the Knowledge Base Refinement dataset. The introduction of similarities (via duplication or segment copying) and contradictions (via targeted factual modification and insertion) was a controlled, deterministic part of the generation pipeline. Consequently, the ground truth identifying these problematic article pairs is known by construction, stemming directly from these explicit generation steps.

## D HUMAN EVALUATION FOR REALISM

To assess the perceived realism of our generated data, we conducted a human evaluation study involving domain experts. We presented three validators with 1000 randomly sampled conversations generated by our pipeline. The experts were given the following instruction and were not informed about the synthetic nature of the data: *"You are given 1000 conversations for our partner GreenBuild, some of which are real conversations collected from the platform and some of them are generated from ChatGPT. Please classify them in Real/Fake category and share insights/reason for your answer."*

The results of the evaluation are summarized below:

- **Validator 1:** Checked 430 conversations, marking 240 as "Confidently Real."
- **Validator 2:** Checked 190 conversations, marking 120 as "Confidently Real."
- **Validator 3:** Checked 380 conversations, marking 200 as "Confidently Real."

Overall, industry experts classified 56% of the synthetic conversations as real. An analysis of the conversations marked as "fake" revealed that the classification was often due to subtle artifacts of the generation process rather than fundamental flaws in conversational flow. Common reasons provided by the experts included:

- *"At the end, after the supervisor was notified to assist, the continued conversation does not make any sense."*
- *"[Customer Address] is mentioned by customer instead of giving address which seems like gpt generated."*
- *"customer gave the mail as "david@example.com" which seems fake to me."*

These findings highlight specific nuances that arose from large-scale generation, such as the use of placeholder data and logical inconsistencies in complex conversational branches. We acknowledge this scope for improvement and plan to enhance the conversation simulation models in future versions of the dataset as we expand to more domains.

## E  BENCHMARK EVALUATION PROCEDURES

This appendix details the methodologies for evaluating models against the five benchmark tasks within CXMArena, guiding researchers on utilizing the task-specific datasets for their experiments. The code for benchmark evaluation can be found at [link redacted for review].

### E.1  KNOWLEDGE BASE REFINEMENT

To evaluate performance on Knowledge Base Refinement, models are tested using the dataset component containing labeled article pairs. The evaluation assesses a model's ability to classify the relationship between two articles as 'similar', 'contradictory', or 'unrelated'. Following the baseline approach described in Section 5.1, one common method involves generating vector embeddings for each article in a pair and calculating a similarity score (e.g., cosine similarity), which can then be thresholded for classification. Performance is measured against the ground truth labels using standard classification metrics: Precision, Recall, and F1-score, with a focus on correctly identifying 'similar' and 'contradictory' pairs. Researchers may also explore alternative methods, such as fine-tuning classifiers or employing large context window models for direct comparison.

### E.2  INTENT TAXONOMY DISCOVERY AND PREDICTION

Evaluating Intent Prediction requires the dataset component that paris conversation transcripts with ground truth intent labels and the associated taxonomy definition. Models process a full conversation transcript to predict a single intent label from the provided taxonomy. The primary evaluation metric, as used in the baselines (Section 5.2), is Accuracy, determined by an exact match between the model's prediction and the ground truth label. For a more nuanced assessment of semantic correctness, particularly when exact labels might differ but meaning aligns within the taxonomy, LLM-based verification can be employed, potentially using a prompt structure like that conceptualized in Figure 11.

### E.3  AGENT QUALITY ADHERENCE

For the Agent Quality Adherence task, evaluation uses the dataset comprising conversation transcripts, specific quality assessment queries, and ground truth Boolean answers. Models are given a transcript and a query, and are primarily evaluated on their ability to produce the correct Boolean (True/False) answer. This comparison against the ground truth forms the basis for calculating Accuracy, reported at both the Question Level and Case Level as in the baselines (Section 5.3). LLM-based models are typically used for this task, guided by prompts incorporating the transcript and query (conceptualized

**Intent Prediction**

**Objective:** Analyze the provided summary of a customer care call to determine the most applicable Contact Driver Intent from an established intent taxonomy.

**Definition**: *Contact Driver Intent* encapsulates the initial concern or inquiry made by a customer, prompting them to reach out to customer support. This intent could directly highlight an issue, suggest a more complex underlying problem, or inquire about a specific aspect of the service or product. It must align with the provided intent taxonomy.

**Instructions**:
 1. Comprehend: Thoroughly read the call summary to understand the customer's primary concern or query.
 2. Identify: Determine the Contact Driver Intent that best matches the summary. This intent should precisely reflect the main reason behind the customer's call, as evident from the summary.
 3. Explain and Classify* Provide a succinct explanation for selecting this particular intent, ensuring your reasoning is clear and directly related to elements mentioned in the call summary.
 4. Fallback Intent: If the summary's content does not fit any intent within the taxonomy precisely, you should choose "Others" as the Contact Driver Intent.

**Input:**
 – Call Summary:
 *[...Hide details for space...]*

 – Universal Intent Taxonomy:
 *[...Hide details for space...]*

Required Output Format: Structure your output as a proper json with keys and values in double quotes in the following form:

{{
 "Intent explanation": "<Reasoning behind the chosen intent>",
 "Intent": "<Identified contact driver intent>"
}}

**Note:**
 1. Ensure that you are predicting the contact driver intent i.e. the initial concern or enquiry made by the customer.
 2. The predicted contact driver intent should be present in the given intent taxonomy.
 3. Predict only one contact driver intent which fits the situation perfectly.
 4. Keys and values in Output json should be in proper double quotes. [IMPORTANT]
 5. DO NOT Generate any additional Notes or explanation. [IMPORTANT]

Figure 11: Structure of a prompt for LLM-based evaluation of Intent Prediction.

in Figure 12). Assessing any additionally provided evidence (like message IDs) would necessitate further evaluation steps beyond the baseline Boolean accuracy.

**Agent Quality Adherence**

**Instructions:**
Given the following conversation and questions, answer the questions in below mentioned list of json format in English language

**Sample response format:**
 [{
   "Question": "Question",
   "Explanation": "Explanation",
   "Answer": "Boolean Answer",
 },
 {
   "Question": "Question",
   "Explanation": "Explanation",
   "Answer": "Boolean Answer",
 }]

Output must follow above json format. Response field definitions are given below:

 1. Question: Question for which answer is given
 2. Answer: Answer the given question as "Yes" or "No"
 3. Explanation: Provide a detailed explanation for your answer. Explanation must be in english language

Read the following conversation between customer and agent carefully and given question and descriptions. Answer the question in the json format:

 **Conversation:**
 *[...Hide details for space...]*

 **List of Questions:**
 *[...Hide details for space...]*

Figure 12: Structure of a prompt for LLM-based evaluation of Agent Quality Adherence.

### E.4 ARTICLE SEARCH

Article Search evaluation utilizes the query-article pairs dataset alongside the full Information KB as the search corpus. Evaluation can target either the retrieval accuracy or the quality of generated answers in a RAG pipeline. For retrieval-focused evaluation, models predict a ranked list of KB article identifiers for a given query; performance is measured using standard IR metrics like Recall@k, Precision@k, or MRR against the ground truth article(s).

### E.5 MULTI-TURN RAG

Evaluating Multi-turn RAG involves the dataset of conversation contexts paired with ground truth KB passage identifiers and the full KB. Models process the conversation history up to the user's last utterance to predict a ranked list of KB passages or articles relevant for the agent's subsequent turn. The primary evaluation method assesses retrieval effectiveness using standard metrics, predominantly Recall@k (as reported in Section 5.5), by checking if the model's top k predictions include the ground truth passage identifier(s) used in the simulation. Some methodologies might involve an intermediate step where an LLM generates a search query from the conversation context (conceptual prompt in Figure 13) before the retrieval system ranks passages.

**Multi-turn RAG**

**Role:** You are an AI assistant tasked with helping a customer support agent find relevant information quickly.

**Context:** You will be given a snippet from an ongoing conversation between a customer and a brand agent. The agent needs to find internal documentation (such as FAQs, knowledge base articles, troubleshooting guides, or policy details) to address the customer's latest point.

**Task:** Analyze the provided conversation snippet, paying close attention to the **customer's most recent**

**message(s)**. Identify the core issue, question, or request being presented. Extract key entities (e.g., product names, error codes, specific features, account details, policy names) and the customer's implied intent.

**Goal:** Generate a concise, keyword-focused search query that the agent can use directly in an internal knowledge base or search system to find the most relevant documents for formulating their *next* response. The query should prioritize the immediate problem or question raised by the customer.

**Conversation Snippet:**
 *[...Hide details for space...]*

**Output Instructions:**
• Generate **only** the search query.
• Do not include any introductory phrases, explanations, or labels (like "Query:").
• The query should be optimized for keyword-based search retrieval.

Output Query:

Figure 13: Structure of a prompt for LLM-based evaluation of Multi-turn RAG query generation.

## F COST AND LATENCY ANALYSIS

All LLM-based evaluations in our benchmarks were performed using the **Gemini-2.0-Flash** model. The associated costs were calculated based on the following API pricing: $0.01 per 1 million input tokens and $0.04 per 1 million output tokens. A detailed breakdown of inference cost and latency for each task is presented in Table 10.

Table 10: Inference cost and latency benchmarks for LLM-based tasks using Gemini-2.0-Flash.

| Task | Records Evaluated | Avg. Cost per Record ($) | Total Cost ($) | Avg. Output Tokens | Avg. Latency (s) |
|---|---|---|---|---|---|
| Multi-Turn RAG Query Gen. | 566 | 0.00011 | 0.063 | 28 | 0.72 |
| Multi-Turn RAG Response Gen. | 566 | 0.00120 | 0.660 | 69 | 1.31 |
| Tool Selection | 456 | 0.00120 | 0.540 | 43 | 1.34 |
| Agent Quality Monitoring (AQM) | 5,199 | 0.00029 | 1.500 | 327 | 2.80 |
| Intent Prediction | 979 | 0.00099 | 0.970 | 78 | 1.38 |

  • The total commercial inference expense for evaluating all LLM-based tasks in the benchmark is **$3.73**.

- There is a strong positive correlation between the average number of output tokens and the mean LLM response latency. Tasks requiring longer generated outputs, such as Agent Quality Monitoring, consistently exhibited higher response times.

- The reported mean latencies correspond to the LLM's response time for a single input. The overall wall-clock time for processing the entire dataset is shorter due to parallelized inference.

- The Knowledge Base Refinement task involved no commercial API costs, as its evaluation was performed entirely using open-source sentence embedding models, as detailed in our provided evaluation code.

## G   ERROR AND VARIABILITY ANALYSIS

To provide insight into the stability and variability of our baseline results, we conducted an analysis on a randomly sampled subset of 100 datapoints for each primary task. The following results report the mean metric alongside the 95% confidence interval (CI), illustrating the expected range of performance.

- **Agent Quality Adherence:** Conversation-level accuracy was 0.32 ($\pm$0.075), while the more granular question-level accuracy reached 0.86 ($\pm$0.020).

- **Knowledge Base Refinement:**
  - For identifying *contradictory* pairs, precision was 0.02, recall was 0.45, and the F1 score was 0.03.
  - For identifying *similar* pairs, precision was 0.04, recall was 0.59, and the F1 score was 0.07.
  - Confidence intervals were $\pm$0 for all metrics, as the sentence embedding models used for this task produce deterministic outputs.

- **Intent Prediction:**
  - Taxonomy A: 0.29 ($\pm$0.072) accuracy.
  - Taxonomy B: 0.58 ($\pm$0.014) accuracy.
  - Taxonomy C: 0.14 ($\pm$0.000) accuracy.
  - The results show higher variability for Taxonomy A, while Taxonomy C's performance was completely non-varying on this sample.

- **Multi-turn RAG (Response Generation Outcomes):** The proportion for each outcome category was:
  - Correct: 0.65 ($\pm$0.020)
  - Incorrect: 0.14 ($\pm$0.015)
  - Hallucinated: 0.06 ($\pm$0.022)
  - Refusal: 0.15 ($\pm$0.026)

- **Tool-Calling Accuracy:** The accuracy for tool selection varied with the number of available tool candidates:
  - 16 candidates: 0.44 ($\pm$0.014)
  - 32 candidates: 0.46 ($\pm$0.052)
  - 64 candidates: 0.43 ($\pm$0.000)
  - 96 candidates: 0.40 ($\pm$0.025)
  - 128 candidates: 0.38 ($\pm$0.052)
  - Confidence intervals were minimal for 16 and 64 candidates, with higher variability observed as the number of tool choices increased.

## H   MULTILINGUAL AND CROSS-DOMAIN GENERALIZATION

To validate the generalizability of our data generation pipeline beyond the primary English-language benchmark, we created two additional datasets in different languages and business domains:

**French/Luxury Cosmetics** and **German/Smartphones**. This appendix provides summary statistics (Table 11) and a consolidated overview of baseline performance across the benchmark tasks for these new datasets (Table 12). The generation and evaluation methodologies remained consistent with the primary English benchmark.

Table 11: Key statistics for the multilingual datasets.

| Metric | French/Luxury Cosmetics | German/Smartphones |
|---|---|---|
| Total Conversations | 997 | 499 |
| KB Articles | 2435 | 2347 |
| Vocabulary Size | 8364 | 6604 |
| Mean Turns per Conversation | 24 | 23 |

Table 12: Consolidated baseline performance on multilingual datasets. Results are reported as `mean ± 95% CI` and show the score for a representative metric from the top-performing model/pipeline for each task.

| Task | Metric | French/Luxury Cosmetics | German/Smartphones |
|---|---|---|---|
| Knowledge Base Refinement | F1 Score (Similarity) | $0.074 \pm 0.000$ | $0.074 \pm 0.000$ |
| Intent Prediction | Accuracy | $0.155 \pm 0.009$ | $0.567 \pm 0.087$ |
| Agent Quality Adherence | Question Level Accuracy | $0.630 \pm 0.017$ | $0.517 \pm 0.021$ |
| Multi-turn RAG | Correctness Rate | $0.700 \pm 0.068$ | $0.567 \pm 0.100$ |
| Tool-calling | Accuracy (128 tools) | $0.232 \pm 0.015$ | $0.237 \pm 0.029$ |

The performance across these datasets confirms that the operational challenges observed in the English benchmark persist in different linguistic and domain contexts. Notably, the low F1 scores for Knowledge Base Refinement and the difficulty of tool-calling in a large candidate pool remain consistent challenges. These findings underscore the robustness and utility of our generation pipeline for creating diverse and challenging CXM evaluation scenarios that can drive future research in multilingual and cross-domain AI applications.

## SUPPLEMENTARY MATERIALS

All resources, including the datasets and the source code for our generation and evaluation pipelines, are provided as supplementary materials to ensure full reproducibility and to facilitate future research. The components are detailed below.

### PUBLICLY AVAILABLE DATASETS

The CXMArena benchmark suite is available on Hugging Face in three languages.

- **English Dataset**: CXMArena
  https://huggingface.co/datasets/tempuser1291480124/CXMArena

- **German Dataset**: CXMArenaGerman
  https://huggingface.co/datasets/tempuser1291480124/
  CXMArenaGerman

- **French Dataset**: CXMArenaFrench
  https://huggingface.co/datasets/tempuser1291480124/
  CXMArenaFrench

### SOURCE CODE

The complete source code is submitted as a supplementary ZIP file, organized into two main components as described in the project's README file.

- data_builder/: A modular and scalable pipeline for generating synthetic, brand-specific CXM datasets that simulate real-world business scenarios, including knowledge bases, customer conversations, and operational tasks.

- evaluator/: A comprehensive evaluation framework designed to benchmark AI models on the generated datasets across all five core CXM tasks: Knowledge Base Refinement, Intent Prediction, Agent Quality Adherence, Article Search, and Multi-turn RAG with Integrated Tools.

