# OpenReview forum: "CXMArena: Unified Dataset to benchmark performance in realistic CXM Scenarios"
_ICLR.cc/2026/Conference — Submitted to ICLR 2026_

### Official Review · Reviewer_nXhr · 2025-10-19

**Soundness:** 1
**Presentation:** 2
**Contribution:** 2
**Rating:** 2
**Confidence:** 4

**Summary:**

The authors developed a new way to automatically generate a complete, brand-specific Knowledge Base (KB) for Customer Experience Management (CXM) problems. They have publicly released the entire dataset, which includes articles, synthesized user queries, and tool definitions.

The paper argues that existing benchmarks fall short for CXM because they focus on general, open-domain knowledge and generic tools. This work tackles that gap by creating a benchmark that includes a full suite of core tasks relevant to CXM, which requires specialized brand/product knowledge and the use of tailored tools, just like in real-world customer support scenarios.

Using their synthetic KB, the authors designed and benchmarked a suite of tasks to evaluate different LLM capabilities. This includes an Intent Prediction task for classification of user needs, an Article Retrieval task for retrieval of relevant documents, a Tool Use task for executing API calls, and a Quality Adherence task to evaluate if responses follow brand guidelines. The benchmark also includes an end-to-end Multi-turn RAG with Tool Use task, which is a generation problem that integrates all these skills to produce a final, context-aware response.

**Strengths:**

The paper identifies a critical gap in existing benchmarks, which typically focus on open-domain tasks and fail to address the unique challenges of Customer Experience Management (CXM), such as specialized knowledge retrieval, tailored tool use, and quality adherence.

A key contribution is the novel data synthesis pipeline that generates a complete and realistic brand-specific Knowledge Base. Synthesizing such data is valuable & practical as it enables the creation of a rich CXM environment for research without exposing sensitive company data or compromising user privacy.

The work provides a comprehensive benchmark by designing and evaluating performance on a suite of diverse tasks relevant to CXM, including retrieval, tool use, and end-to-end multi-turn RAG, thereby establishing important baselines for future research.

**Weaknesses:**

The benchmark's primary weakness is a lack of a clear rationale for performing several isolated, synthesized tasks on a synthesized knowledge base. For example, both the article retrieval and tool-calling tasks are designed in isolation. While this is a valid approach, the paper does not explain the unique advantages of this synthetic setup over existing approaches that synthesize search queries, or tool calls from openly available articles and real-world tools (e.g. ToolBench).  A stronger justification would help frame the benchmark as a cohesive suite of tasks designed to address novel CXM challenges, rather than a collection of disconnected components. (e.g. any unique challenge around jointly performing KB retrieval, tool call during a multi-turn RAG problem?)

While the paper does involve an evaluation on an end-to-end RAG task, a key methodological detail that remains unclear is how tool outputs are simulated during the multi-turn RAG evaluation, or if tool calls are simulated at all. In addition, while the paper mentions llm-as-a-judge is used to evaluate the final multi-turn RAG output, it is not clear if the evaluation is grounded in the golden response or the correct sources.

**Questions:**

In the multi-turn RAG evaluation, how are tool outputs simulated and provided to the model after a tool call is made?

For the LLM-as-a-judge evaluation, is the judge grounded against the knowledge base or golden responses to verify factual accuracy?

It seems that KB_3211 is referenced in the multi-turn set but is missing from the articles subset. Could you clarify this data discrepancy?

---

### Official Review · Reviewer_wsXP · 2025-10-21

**Soundness:** 2
**Presentation:** 3
**Contribution:** 2
**Rating:** 2
**Confidence:** 4

**Summary:**

The authors introduce CXMArena: a novel, large-scale synthetic benchmark dataset specifically designed for evaluating AI in operational Customer Experience Management contexts.

**Strengths:**

- Real-world distribution because of controlled noise injection (simulated ASR errors, interaction fragments) from SMEs and rigorous automated validation.

- Authors introduce five tasks: Knowledge Base Refinement, Intent Prediction, Agent Quality Adherence, Article Search, and Multi-turn RAG with Integrated Tools.

- Pipeline applied to different domains and languages.

- The authors introduce a pipeline to synthetically generate the knowledge base specific to a fictional brand and then uses these KBs to generate realistic conversations along with noise.

**Weaknesses:**

- My main concern is on the correctness of the synthetic data using an LLM (Gemini in this case) and the LLM as a judge evaluation without a human in the loop.

- Contradiction detection baseline would be very insightful since this is one of the tasks needed for Knowledge Base refinement.

- Do you generate all this dataset synthetically given a seed prompt about the brand name and its type? I'm not convinced how you can ensure highly fidelity data since there is no human in the loop. How do you avoid hallucinations? For instance for the KB refinement task how do you know that similar KBs are correctly labelled?

- GPT 4o as a judge and no human evaluation.

- LLM was used for checking the dataset correctness with no human expert in the loop.

**Questions:**

In the abstract you mention "The entities closely represent real-world distribution because of controlled
noise injection (informed by domain experts)". What kind of information do the experts provide?

---

### Official Review · Reviewer_52hR · 2025-10-26

**Soundness:** 2
**Presentation:** 3
**Contribution:** 2
**Rating:** 4
**Confidence:** 4

**Summary:**

The authors propose a new dataset for evaluation of LLM response in Customer Service. They tackle important problems like Knowledge base based questions, tool calling, multi-turn question answers etc. They provide limitations of current datasets and propose a system which can address these limitations.

**Strengths:**

The authors bring forth a very important problem and one which practitioners constantly face. One very important aspect of public datasets and papers is that they very rarely to industry situations and the authors point it out very well. They also do a comprehensive study of the different challenges situations in Customer Service via LLMs in Section 2 and bring about limitations of exisiting datasets very well.

**Weaknesses:**

The authors highlight that most public datasets have limitations. This is a very valid concern and they have given significant citations establishing the limitations of existing datasets in Section 2. What is not clear is how is this alleviated in their work. Looking at one example they mention "We simulate real-world data quality issues by introducing controlled redundant and contradictory information from one article to another, creating data for developing KB maintenance techniques" - it is not clear how is this going to solve the very real problems in lines 088-103.  Similarly, their description on Multi-turn RAG in lines 252-257 do not explain how the limitations discussed in Section 2 are addressed. The paper largely glosses over details (some information is present in Appendix but that is not under detailed review) and more importantly even from Appendix it is not clear how the concerns are addressed.

Next the authors provide multiple evaluation results with different models however it is hard to understand if this is good or bad without a benchmark. They provide some examples in the appendix however they do not give examples of how they are different from existing datasets making an evaluation of the marginal improvement challenging.

**Questions:**

Please address the concerns in the weaknesses section above.

---

### Official Review · Reviewer_gNnY · 2025-11-01

**Soundness:** 3
**Presentation:** 3
**Contribution:** 3
**Rating:** 6
**Confidence:** 3

**Summary:**

1. CXMArena manuscript introduces a unified, large-scale which is basically synthetic benchmark for evaluating AI models in Customer Experience Management (CXM). The synthetic data benchmark will be used in many real world scenarios

2. The manuscript tells about five operational tasks: Knowledge Base Refinement, Intent Prediction, Agent Quality Adherence, Article Search, and Multi-turn RAG with Integrated Tools which are all pretty well explained and in depth.

3. All the dataset which are created using a scalable and high value LLM-powered pipeline that simulates realistic customer-agent interactions with controlled noise injection for authenticity. The LLMs used are pretty in depth with metrics supported. The synthetic data can be used in very critical in CXM scenarios.

**Strengths:**

1. Comprehensive benchmark covering core CXM tasks beyond usual fluency. The CXM tasks usually have real word scenario use cases so the manuscript is useful.

2. Synthetic yet realistic data with strong alignment to real-world metrics. The realistic nature of this data has very strong alignment and large scale value in very in depth metrics.

3. Cross-domain and multilingual support (English, French, German). The cross domain languages are very well used in here.

4. Provides baseline results for multiple models. Multiple models are been used like GPT and others.

**Weaknesses:**

1. Although the manuscript is being used for synthetic, it may miss subtle nuances of real human behavior. Also LLM with human as a judge could be helpful to explore that might strengthen the findings
2. Dependent on biases and limitations of LLMs used for generation.
3. Currently focused on one domain with limited real-world diversity. Also multi domain alignment could be helpful

**Questions:**

1. How will CXMArena handle continuous domain drift in real CXM data? The drift can be measured and how can be used.
2. Can the synthetic pipeline generalize to non-English, multi-brand, or emotion-rich interactions?
3. How could MCP be used and how some performance benchmarks on MCP be used

---

### Meta-Review · Area_Chair_dkiv · 2025-12-25

**Summary:**

The authors do not give the rebuttal. Therefore, the concerns from all reviewers are not addressed. In addition, most of reviewers give negative comments and the 3 of all 4 scores from reviewers are negative.

**Reviewer Concerns:**

The authors do not give the rebuttal. Therefore, the concerns from all reviewers are not addressed.

**Reviewer Scores:**

For the original scores before the rebuttal, 3 of all 4 scores from reviewers are negative. The authors do not give the rebuttal. Therefore, I think all reviewers tend to keep their original scores.

---

### Decision · Program_Chairs · 2026-01-26

Reject